# ENHANCING COMPOSITIONAL TEXT-TO-IMAGE GENERATION WITH RELIABLE RANDOM SEEDS

**Shuangqi Li[1], Hieu Le[1], Jingyi Xu[2], Mathieu Salzmann[1]**
[1]EPFL, Switzerland      [2]Stony Brook University, USA
{shuangqi.li, mathieu.salzmann}@epfl.ch
{hle, jingyixu}@cs.stonybrook.edu

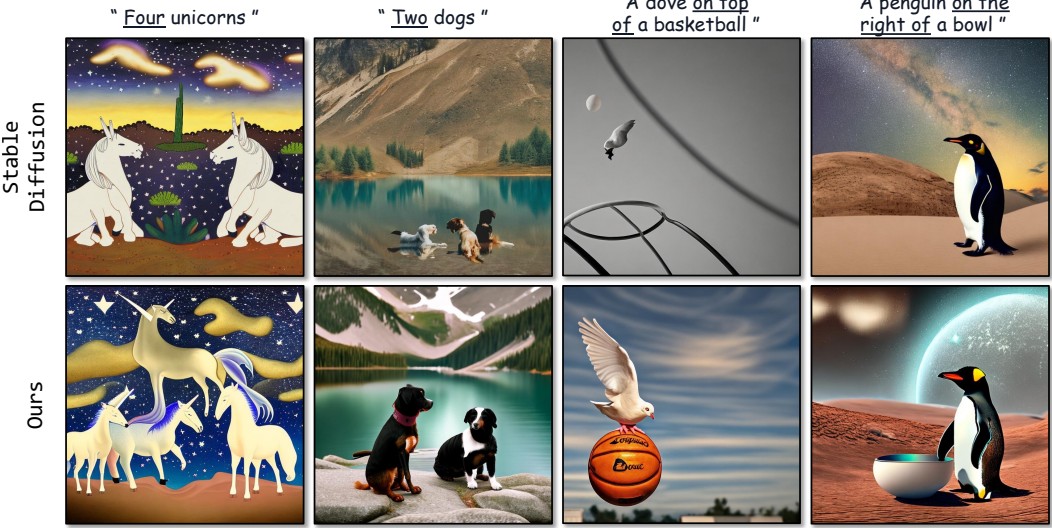

Figure 1: **Example images generated by Stable Diffusion 2.1 and ours.** Existing text-to-image diffusion models are prone to making mistakes at numeracy and spatial relations.

## ABSTRACT

Text-to-image diffusion models have demonstrated remarkable capability in generating realistic images from arbitrary text prompts. However, they often produce inconsistent results for compositional prompts such as "two dogs" or "a penguin on the right of a bowl". Understanding these inconsistencies is crucial for reliable image generation. In this paper, we highlight the significant role of initial noise in these inconsistencies, where certain noise patterns are more reliable for compositional prompts than others. Our analyses reveal that different initial random seeds tend to guide the model to place objects in distinct image areas, potentially adhering to specific patterns of camera angles and image composition associated with the seed. To improve the model's compositional ability, we propose a method for mining these reliable cases, resulting in a curated training set of generated images without requiring any manual annotation. By fine-tuning text-to-image models on these generated images, we significantly enhance their compositional capabilities. For numerical composition, we observe relative increases of 29.3% and 19.5% for Stable Diffusion and PixArt-$\alpha$, respectively. Spatial composition sees even larger gains, with 60.7% for Stable Diffusion and 21.1% for PixArt-$\alpha$. Our code is available at https://github.com/doub7e/Reliable-Random-Seeds.

## 1 INTRODUCTION

Text-to-image synthesis transforms textual descriptions into images, supporting various applications like digital arts, visualization, design, and education. Diffusion models, such as Stable Diffusion (Rombach et al., 2022) and PixArt-$\alpha$ (Chen et al., 2023a), have become the gold standard for generating high-quality, diverse images, far surpassing earlier GAN-based methods (Reed et al.,

2016; Liang et al., 2020; Tao et al., 2022). However, these models are not without limitations: They often struggle with compositional prompts, frequently generating inaccurate numerical and spatial details (*e.g.*, "four unicorns" or "a dove on top of a basketball"), as shown in Figure 1. To remedy this issue, some works improve the models by incorporating layout inputs for object positioning (Dahary et al., 2024; Chen et al., 2023b; Zheng et al., 2023), while others leverage Large Language Models (LLMs) to generate bounding box coordinates from prompts (Qu et al., 2023; Lian et al., 2023; Feng et al., 2024a; Chen et al., 2023c). These approaches mitigate the issue to some extent, but also face various challenges, including labor-intensive layout creation, limited diversity, biases, and potential trade-offs in image realism and generation speed.

Furthermore, the exact reasons for the model's inconsistencies with compositional prompts remain unclear, as most studies focus on understanding the overall performance of the model without addressing specific prompt types. For instance, Guo et al. (2024a) demonstrate that the initial noise significantly influences the quality of generated images, with certain noise patterns leading to semantically coherent outputs while others not. Similarly, Xu et al. (2024) highlight that latent noise vectors located in high-density regions are more likely to yield higher-quality samples. These findings commonly suggest a significant role of noise in understanding the underlying mechanisms of diffusion models.

Our research delves deeper into this relationship, revealing a strong connection between initial noise and the fidelity of generated images for compositional prompts. Specifically, we find that certain noise signals are more likely to result in accurate object counts and positions. This is because certain random seeds tend to guide the model in placing objects within specific areas and patterns, *i.e.*, layouts, and interestingly, some layouts consistently yield more accurate results than others. For example, some seeds lead to a "four-grid" layout, which simplifies the task of placing four distinct objects. Similarly, layouts that arrange objects vertically are more likely to generate accurate spatial relationships, such as "*on top of.*" Conversely, layouts that cluster objects into small regions often result in missing objects and perform poorly on any compositional task. In fact, we observe a 6% accuracy improvement for both Stable Diffusion and PixArt-$\alpha$ when simply using these reliable seeds instead of uniformly randomizing.

Furthermore, these reliable seeds enable us to curate a set of images with relatively accurate object counts and placements in an automatic manner. Fine-tuning text-to-image models on these images significantly enhance the models' compositional capabilities. For numerical composition, we observe relative increases of 29.3% and 19.5% for Stable Diffusion and PixArt-$\alpha$, respectively. Spatial composition sees even larger gains, with 60.7% for Stable Diffusion and 21.1% for PixArt-$\alpha$.

In summary, our work identifies the critical role of initial noise in the generation of accurate compositions, offering a new perspective on how text-to-image models can be improved for complex prompts. By leveraging reliable noise patterns, we achieve significant improvements in both numerical and spatial composition without the need for explicit layout inputs.

## 2 BACKGROUND AND RELATED WORK

**Text-to-image Diffusion Models.** Diffusion models have emerged as a prominent class of generative models, renowned for their ability to create highly detailed and varied images, especially in a text-to-image fashion (Ramesh et al., 2022; Saharia et al., 2022; Rombach et al., 2022). These models integrate pretrained text encoders, such as CLIP (Radford et al., 2021) and T5 (Raffel et al., 2020), to enable the understanding of textual descriptions, and pass the encoded textual information to cross-attention modules. Cross-attention is the key component for most text-to-image models, such as Stable Diffusion Rombach et al. (2022), to guide the visual outputs with textual information by dynamically focusing on the relevant encoded information when generating corresponding image pixels. However, despite being effective in many aspects of image generation including style and content, these models frequently fail to resolve finer details in the text prompts, such as numeracy, spatial relationships, negation, attributes binding, etc. (Huang et al., 2023)

**Initial Noise of Diffusion Sampling.** The initial noise of the sampling process is known to have a significant impact on the generated image for diffusion models, and is usually randomly drawn from a Gaussian distribution with a random seed to generate diversified images. Lin et al. (2024); Guttenberg (2023); Everaert et al. (2024) found that the initial noise contains some low-frequency

information such as brightness, and Guttenberg (2023) proposed to offset the noise to address diffusion model's inability to generate very bright or dark images. Previous works have also attempted to optimize the initial noise to produce images that align better with the text prompts (Guo et al., 2024b) or follow a given layout (Mao et al., 2023). Additionally, Samuel et al. (2024) demonstrate that carefully selected seeds can help generate rare concepts for Stable Diffusion.

**Enhancing Compositional Text-to-Image Generation.** An important line of research attempts to enhance the compositional generation ability of text-to-image diffusion models at inference time by dividing the generation process into two stages: Text-to-layout and layout-to-image. Automatic text-to-layout can be achieved by querying LLMs with carefully-crafted prompts (Qu et al., 2023; Lian et al., 2023; Feng et al., 2024a; Chen et al., 2023c). Layout-to-image usually enforces the attention maps to follow the provided bounding boxes or masks (Dahary et al., 2024; Chen et al., 2023b; Zheng et al., 2023; Bar-Tal et al., 2023). Attend-and-Excite (Chefer et al., 2023), on the other hand, directly guide the model to refine the cross-attention scores to subject tokens. Another considerable line of work suggest fine-tuning on a smaller, curated dataset to improve text-to-image alignment (Podell et al., 2023; Dai et al., 2023; Segalis et al., 2023). GORS (Huang et al., 2023), proposes to fine-tune on curated generated data weighted by an ensemble metric of three vision-language models assessing attribute binding, spatial relationship, and overall alignment respectively.

Unlike the above, we propose (1) a training-free, extra-computation-free sampling method that exploits reliable seeds for enhanced compositional generation, and (2) a fine-tuning strategy that utilizes self-generated data that are produced with reliable seeds.

## 3 Understanding Initial Seeds For Compositional Generation

In this section, we examine the role of initial seeds in compositional generation. These seeds determine the noise input for diffusion models during image generation. We address three key questions: (1) How do seeds influence object arrangement? (2) Does arrangement correlate with prompt accuracy? (3) Do seeds affect the likelihood of correct image generation?

### 3.1 Initial Seeds Affect Object Arrangements

We find a strong correlation between initial seeds and object arrangement. Note that object arrangement refers to patterns of object placement in terms of relative positions rather than explicit layouts, such as bounding boxes or region masks, which specify the positions, sizes, and shapes of objects in the image (Chen et al., 2023b; Qu et al., 2023; Zheng et al., 2023; Couairon et al., 2023; Jia et al., 2024; Lian et al., 2023; Xu et al., 2023). To evidence this, we used Stable Diffusion 2.1 to generate 512 images based on 8 seeds, 8 classes, and 8 backgrounds, using prompts in the format "Four {*object category*}, {*background setting*}" (see Appendix A.1.4). Figure 2 shows the average cross-attention map corresponding to the object token for each seed. As can be seen, each initial seed leads to distinct areas where objects are more likely to be placed.

We provide visual examples in Figure 3 for two seeds: 50 and 23. For seed 50, it can be seen that the objects tend to be placed in the center diagonally side by side, providing generally more space for the objects. In contrast, for seed 23, objects often occupy very small areas (e.g., the first two images in the first row) and frequently fail to properly appear in the image. Although not perfect, using seed 50 is generally more effective than using seed 23 in guiding Stable Diffusion to generate accurate images for compositional prompts. In fact, for seed 50, 14 out of 16 images are correct, whereas for seed 23, only 3 images are correct. These distinct patterns highlight the significant impact of initial seeds on the overall object arrangement.

### 3.2 Object Arrangements Correlate with Compositional Correctness

To quantitatively investigate the correlation between object arrangement and correct rendering, we used Stable Diffusion 2.1 to generate 300 images for four prompts describing compositional scenes such as "four coins" or "two boats" with random seeds, resulting in 1200 images in total. In Figure 4, we categorize these images based on whether the composition is correctly rendered and compute their average cross-attention maps. Additionally, we visualize the cross-attention maps of the generated images in a 2D plot using t-SNE (Van der Maaten & Hinton, 2008).

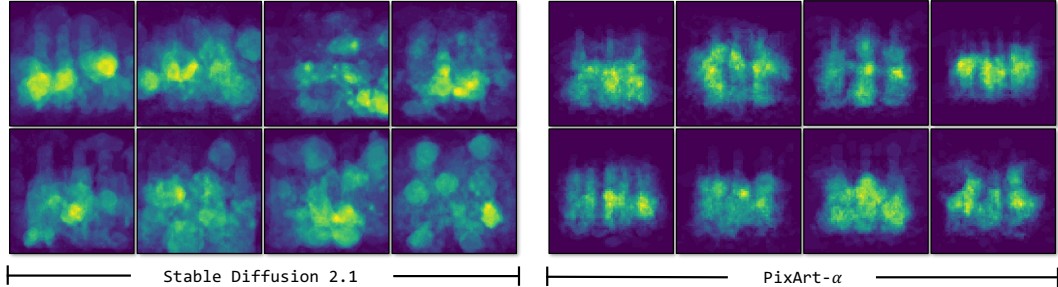

Figure 2: **Initial Seeds and the average attention maps of object tokens.** We generate 64 images for each initial seed from $0 \sim 7$, using Stable Diffusion 2.1 (left) and PixArt-$\alpha$ (right) - each image visualizes one seed. For each seed, we show the average binarized cross-attention maps.

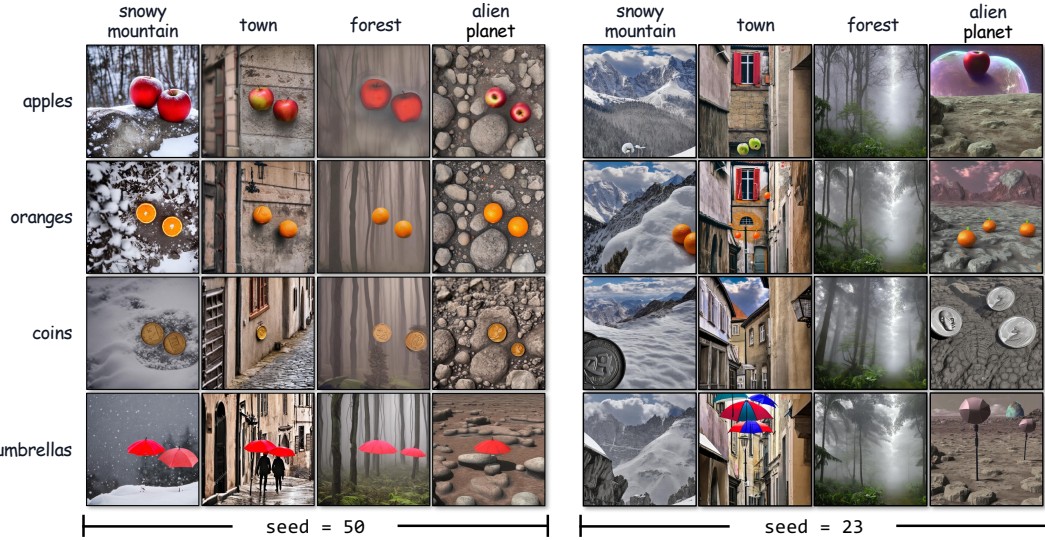

Figure 3: **Initial seeds and object arrangement.** We select two initial seeds to generate images with the prompts "*Two* {object category}, {background setting}", using Stable Diffusion 2.1.

As can be seen, the average cross-attention maps differ significantly between correct and incorrect images. Correct images exhibit consistent image arrangements while incorrect ones do not, suggesting that certain object arrangements strongly benefit compositional generation.

## 3.3 INITIAL SEEDS AFFECT COMPOSITIONAL CORRECTNESS

We now quantify whether initial seeds directly correlate with compositional correctness. To do so, we compare the performance of five candidate seeds. For each seed, we used Stable Diffusion 2.1 (Rombach et al., 2022) to generate 480 images based on compositional instructions (in this case, we asked the model to generate 'four' objects in different contexts) across a set of categories. To evaluate the compositional accuracy of these images, we employed CogVLM2 (Hong et al., 2024), a state-of-the-art vision-language model renowned for its superior performance in visual content understanding and accurate textual description generation. For each generated image, CogVLM2 was queried with prompts such as 'How many apples are in the image?' (see Appendix A.3.1).

Interestingly, the five candidate seeds led to very different model performances: 41.0%, 31.9%, 31.0%, 29.0%, 28.3%, from highest to lowest. A chi-squared test yielded a p-value of $1.2 \times 10^{-4}$, indicating high statistical significance and an extremely low probability of this distribution occurring by chance. Furthermore, we used the same seeds to generate 120 images for objects from unseen categories—different from the categories used above. The top-performing seed achieved 38.3%, significantly outperforming the lowest-performing seed, which only achieved 17.5%. These results suggest that certain initial seeds are more reliable for compositional generation, and their superiority may generalize across object categories and settings.

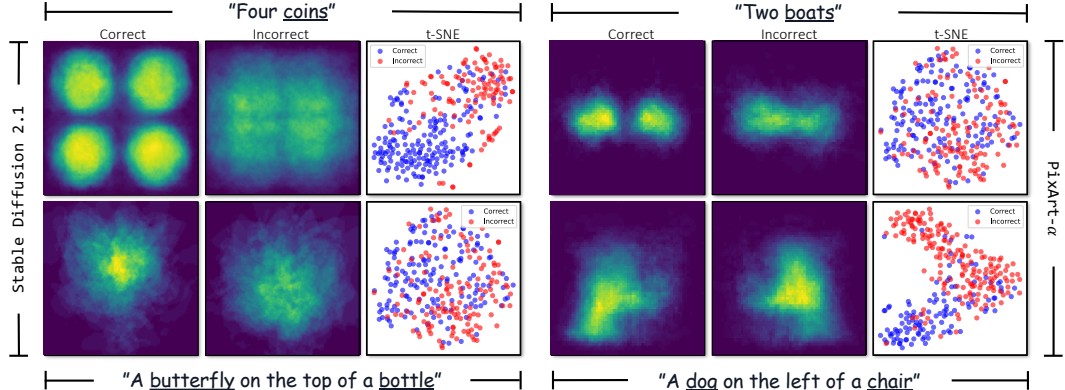

Figure 4: **Averaged object attention masks of generated images with correct and incorrect object counts/positions.** We generate 300 images for each of the four prompts with random seeds, using Stable Diffusion 2.1 (**Left**) and PixArt-$\alpha$ (**Right**). For each prompt, we compute the average of the binarized cross-attention maps. The rightmost plot in each panel visualizes the cross-attention maps of the 300 generated images using t-SNE (Van der Maaten & Hinton, 2008), showing that the attention maps of correct images and incorrect images tend to form different clusters.

## 4 MINING RELIABLE SEEDS FOR ENHANCING GENERATION

Our analyses reveal that some initial seeds lead to significantly better compositional performance. To enhance text-to-image models, a straightforward approach is to use only these seeds for image generation. We go one step further and propose to construct a dataset generated from those reliable seeds and use it to improve the overall model performance via fine-tuning.

To this end, we first construct the Comp90 dataset– a prompt dataset for text-to-image generation and reliable seed mining– containing 3,000 text prompts for numerical composition and 3,200 for spatial composition, spanning 90 popular object categories and 12 background settings. We employ the off-the-shelf vision language model CogVLM2 (Hong et al., 2024) for finding reliable seeds. The overview of the proposed approach is illustrated in Figure 5.

### 4.1 DATASET FOR COMPOSITIONAL TEXT-TO-IMAGE GENERATION: COMP90

We collected a set of 90 categories of different types of objects including foods (such as "apple", "hamburger"), animals (such as "dog", "elephant"), common-in-life objects (such as "camera", "bulb"), across various sizes (such as "ant" and "airplane") and shapes (such as "volleyball" and "spoon"). To form diversified text prompts together with the objects, we hand-crafted 12 distinct background settings, ranging from "in an old European town" to "on a rocky alien planet". We randomly divided them into a training set consisting of 60 categories and 8 settings and a test set of 30 categories and 4 settings. We then created text prompts for numerical composition and spatial composition in the following strategy:

- **Numerical prompts**. For each category, we first created 5 text prompts with the number of objects to be generated varying from 2 to 6, each was then appended with each background setting. For example, "*two apples in an old European town*". This yields a total of 2,400 prompts for training and 600 prompts for testing.

- **Spatial prompts**. We consider four types of spatial relations: "on top of", "on the left of", "on the right of", and "under". For each relation, we first generate a large batch of compositional scene candidates by randomly combine two categories with the relation. For example, "an apple on top of a hamburger". Candidates were then provided to GPT-4o (Achiam et al., 2023; OpenAI) to filter out those describing unreasonable compositions like "a table on top of a bowl". After filtering, each candidate was appended with each background setting to obtain text prompts like "*an apple on top of a hamburger, on a rocky alien planet*". In the end, we obtained 2,560 text prompts for training and 640 prompts for testing. More detail on the dataset is provided in Appendix A.1.

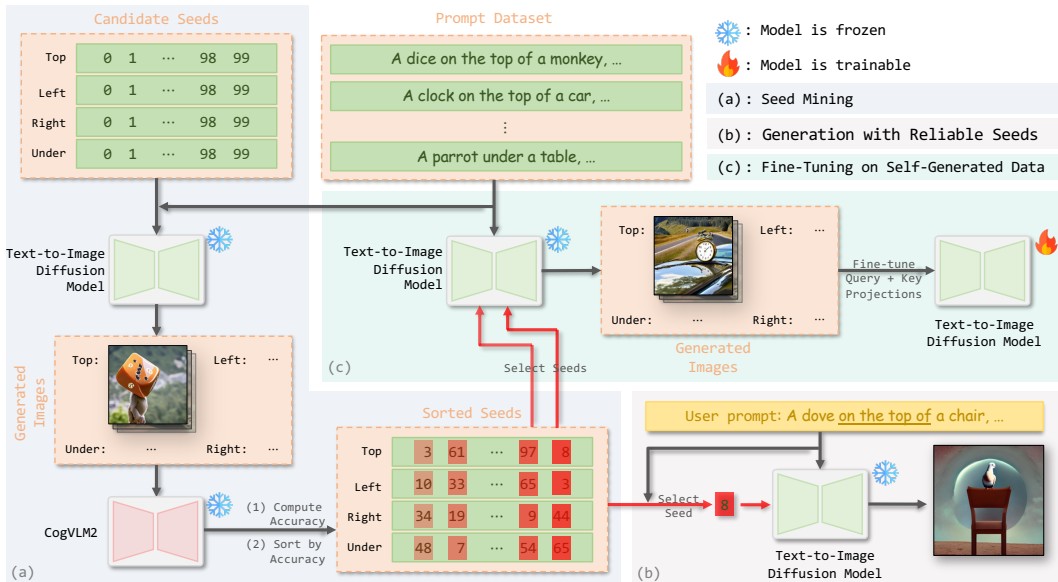

Figure 5: **Overview of the proposed approach.** We take spatial composition as an example to illustrate **(a)** our seed mining strategy. With reliable seeds (*e.g.*, seed 8 in this case), we can **(b)** directly enhance the generation process to improve the compositional accuracy, or **(c)** fine-tune the model to achieve seed-independent enhancement.

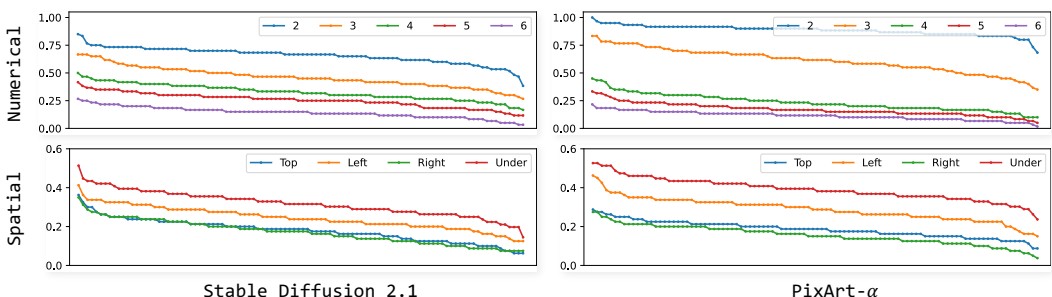

Figure 6: **Accuracy distributions of random seeds on different tasks.** Each line depicts the performance of 100 seeds for the corresponding task, sorted by their performance. As can be seen, top-performing seeds significantly outperform the rest.

## 4.2 RELIABLE SEED MINING

We propose an economical strategy for mining reliable seeds using CogVLM2 and only exploit a small portion of our prompt dataset for this purpose. Our approach considers 100 candidate seeds and proceeds as follows: (1) For each seed, we generate 60 images using prompts composed from the first 15 categories and 4 settings from the training set. This process is repeated 5 times, corresponding to desired object quantities from 2 to 6, generating a total of 30,000 images. (2) CogVLM2 is then prompted to predict the quantity of the queried object in each image. (3) We compare the predicted quantity with the specified quantity. For each possible quantity from 2 to 6, candidate seeds are sorted according to their respective output accuracy. Similarly, for spatial composition, our strategy generates images using text prompts composed from 20 spatial scenes and 4 settings from the training set, producing 80 images per seed per spatial relation. We query CogVLM2 to briefly describe each image first before answering specific questions such as "*Is there an apple on the top of a hamburger in the image?*". For each spatial relation, we sort the candidate seeds according to their output accuracy. More detail is provided in the Appendix A.3.1. Figure 6 illustrates the accuracy distributions of all candidate seeds after sorting.

Table 1: **Comparison of different methods on numerical composition on the Comp90 dataset.** Results for different desired quantities are displayed separately, and "All" denotes the average of all quantities. **Bold** denotes the best value in each column.

| Method | All | | 2 | 3 | 4 | 5 | 6 |
|---|---|---|---|---|---|---|---|
| | Acc↑ | MAE↓ | Acc↑ | Acc↑ | Acc↑ | Acc↑ | Acc↑ |
| Stable Diffusion 2.1 | 37.5 | 1.39 | 75.0 | 43.3 | 29.2 | 23.3 | 16.7 |
| + sampling with reliable seeds | 43.0 | 1.29 | 73.3 | 62.5 | 39.2 | 21.7 | 18.3 |
| + fine-tuning (random) | 41.8 | 1.17 | 70.8 | 55.8 | 39.2 | 21.7 | 21.7 |
| + fine-tuning (reliable) | 48.5 | 1.04 | 85.8 | 63.3 | 32.5 | 33.3 | 27.5 |
| + fine-tuning (random + rectified) | 49.2 | 1.02 | 81.7 | 63.3 | **42.5** | 34.2 | 24.2 |
| + fine-tuning (reliable + rectified) | **51.3** | **0.80** | **86.7** | **65.0** | 37.5 | **37.5** | **30.0** |
| PixArt-$\alpha$ | 34.3 | 1.74 | 76.7 | 50.0 | 18.3 | 18.3 | 7.5 |
| + sampling with reliable seeds | 40.3 | 1.60 | 80.8 | 61.7 | **27.5** | 21.7 | 10.0 |
| + fine-tuning (random) | 35.0 | 1.70 | 77.5 | 51.7 | 17.5 | 21.7 | 6.7 |
| + fine-tuning (reliable) | 41.0 | 1.46 | **85.8** | **68.3** | 20.8 | 20.0 | 10.0 |
| + fine-tuning (random + rectified) | 38.2 | 1.49 | 79.2 | 65.0 | 18.3 | 20.8 | 7.5 |
| + fine-tuning (reliable + rectified) | **41.2** | **1.38** | 83.3 | 65.0 | 20.0 | **22.5** | **15.0** |

### 4.3 FINE-TUNING ON SELF-GENERATED DATA

Solely using reliable seeds for generation would enhance the model performance. However, it inevitably reduces output diversity as users would be limited to a finite number of seed choices for a given prompt. To address this limitation, we propose fine-tuning the model on data generated by itself to generalize the seed-associated behaviors. Using the top-performing seeds for each quantity from 2 to 6, respectively, we generate one image for each of the text prompts in the training set, resulting in an image dataset containing 2,400 pairs of image and prompt. Similarly, for spatial composition, we generate a dataset of 2,560 examples using only the top-performing seeds for each spatial relation respectively. For each generated image, we use CogVLM2 to check its correctness.

Fine-tuning a model on self-generated data should however be handled carefully due to the potential exacerbation of visual biases existing in the generated data (Shumailov et al., 2023). To retain the model's capability as much as possible and effectively enhance the targeted ability of compositional generation, it is essential to fine-tune only the relevant parts of the model and keep other parameters unchanged. Recognizing that image arrangements are significantly correlated with the attention maps (Hertz et al., 2022; Ma et al., 2024; Chen et al., 2024), we opt for only fine-tuning the query and key projection layers in the attention modules. In Appendix A.5.1, we compare this strategy with other parameter choices and demonstrate that our approach leads to the largest improvement while maintaining image quality after fine-tuning.

## 5 EXPERIMENTS

To demonstrate the effectiveness and generalizability of our approach, we evaluate it on two text-to-image models with distinct architectures, Stable Diffusion 2.1 and PixArt. Our assessment focuses on two key compositional generation tasks: Numerical and spatial composition. Additionally, we evaluate the output diversity and image quality of the proposed methods, and demonstrate their superiority over two inference-time methods, LLM-grounded Diffusion (LMD) (Lian et al., 2023) and MultiDiffusion (Bar-Tal et al., 2023). As described in Section 4.1, we conduct seed-mining on a small proportion of the training set of Comp90, and generate data for fine-tuning on the whole training set. We evaluate all methods on the test set of Comp90, which contains 600 text prompts for numerical composition and 640 prompts for spatial composition. For all experiments requiring reliable seeds, we use the top-3 best-performing seeds based on the results of seed mining, unless otherwise specified. The details of the training configurations are in Appendix A.2.

Table 2: **Comparison of different methods on spatial composition on the Comp90 dataset.** Accuracies for the four spatial relations, "on the top of", "on the left of", "on the right of", and "under", are displayed separately, and "All" denotes the average of all relations. **Bold** denotes the best value in each column.

| Method | All | Top | Left | Right | Under |
|---|---|---|---|---|---|
| Stable Diffusion 2.1 | 17.8 | 23.1 | 18.8 | 18.1 | 11.3 |
| + sampling with reliable seeds | 23.4 | 25.0 | 23.1 | 25.0 | 20.6 |
| + fine-tuning (random) | 22.0 | 28.1 | 20.6 | 27.5 | 11.9 |
| + fine-tuning (reliable) | 28.6 | 36.9 | 28.1 | 29.4 | 20.0 |
| + fine-tuning (random + rectified) | 32.0 | 46.3 | 36.9 | 31.9 | 13.1 |
| + fine-tuning (reliable + rectified) | **36.6** | **48.7** | **40.6** | **33.1** | **23.7** |
| PixArt-$\alpha$ | 22.7 | 31.9 | 25.6 | 21.2 | 11.9 |
| + sampling with reliable seeds | 25.6 | 33.8 | **28.1** | 26.3 | 14.4 |
| + fine-tuning (random) | 23.4 | 38.1 | 24.4 | 20.0 | 11.3 |
| + fine-tuning (reliable) | **27.5** | 41.9 | 25.6 | 26.3 | **16.3** |
| + fine-tuning (random + rectified) | 26.6 | **45.0** | 25.6 | 23.1 | 12.5 |
| + fine-tuning (reliable + rectified) | 27.2 | 43.8 | 22.5 | **28.1** | 14.4 |

## 5.1 IMPROVEMENT ON COMPOSITIONAL GENERATION

**Baselines.** We use the pre-trained Stable Diffusion 2.1 ($768 \times 768$) and PixArt-$\alpha$ ($512 \times 512$) as initial baselines, performing normal sampling with random seeds. To demonstrate the effectiveness of the proposed fine-tuning strategy, we evaluate models fine-tuned on self-generated data produced with random seeds. Additionally, we evaluate models fine-tuned on data generated with random seeds and then rectified by CogVLM2, addressing the potential proposal to use self-generated data with text prompts re-captioned by a vision language model.

**Our Methods.** Our methods leverage reliable seeds obtained from our seed mining strategy. Our simplest approach directly uses the pre-trained models to sample with these reliable seeds. Moreover, we evaluate models fine-tuned on data generated with these reliable seeds. To achieve further improvement and demonstrate compatibility with other potential methods, we evaluate models fine-tuned on the previously generated data after using CogVLM2 to rectify the text prompts - we change the instructed prompt such that it matches the content of the generated image.

### 5.1.1 QUANTITATIVE RESULTS

We present quantitative evaluation results for numerical and spatial composition in Tables 1 and 2. We calculate the output accuracy, the ratio of generated images correctly aligning with the text prompts. For numerical composition evaluation, we also compute the Mean Absolute Error (MAE) between the actual generated quantity and the specified quantity. We employ GPT-4o to determine the actual quantities or spatial relations in generated images (details in Appendix A.3.2).

**Sampling with Reliable Seeds.** We can readily improve compositional generation by providing the sampling process with one of the top-performing seeds identified through our mining strategy. Across all evaluated models and tasks, sampling with top-performing seeds from seed mining yields an accuracy improvement of approximately 3% to 6%. While less substantial than fine-tuning methods, this simple, practical plug-and-play technique incurs no additional computational overhead. Our experiments utilized the top-3 seeds; nevertheless, our ablation studies in Appendix A.5.2 demonstrate that the sampling strategy remains substantially effective for top-$k$ values ranging from $k = 1$ to 50.

**Fine-Tuning on Data Generated with Reliable Seeds.** Our fine-tuning strategy consistently outperforms pre-trained models, the proposed sampling strategy, and the baseline fine-tuning strategy using randomly-seeded data by a substantial margin. While our goal is to enable the use of fine-tuned models with random seeds in standard applications, we discovered that sampling these fine-tuned models with reliable seeds can yield even further performance gains (results in Appendix A.4).

**CogVLM Rectification.** Fine-tuning T2I models on re-captioned images (Podell et al., 2023; Dai et al., 2023; Segalis et al., 2023) is not novel. We compare such methods with ours by fine-tuning on

Table 3: **Comparison of evaluation results obtained by GPT-4o and human.** "-G" stands for GPT-4o evaluation results while "-H" represents human results.

| Method | Numerical | | | | Spatial | |
| --- | --- | --- | --- | --- | --- | --- |
| | Acc-G | Acc-H | MAE-G | MAE-H | Acc-G | Acc-H |
| Stable Diffusion 2.1 | 37.5 | 37.8 | 1.39 | 1.33 | 17.8 | 17.0 |
| + fine-tuning (reliable + rectified) | **51.3** | **52.8** | **0.80** | **0.67** | **36.6** | **38.6** |

Table 4: **Comparison of different methods on the aesthetic score and the recall.** Our method significantly improves the accuracy, while coming at a much lower loss of aesthetic score and recall than the state of the art (LMD, MultiDiffusion and Ranni).

| Method | Numerical | | | Spatial | | |
| --- | --- | --- | --- | --- | --- | --- |
| | Acc↑ | Aes↑ | Rec↑ | Acc↑ | Aes↑ | Rec↑ |
| Stable Diffusion 2.1 (768 × 768) | 37.5 | **5.19** | **76.7** | 17.8 | **5.38** | **70.4** |
| + sampling with reliable seeds | 43.0 | 5.23 | 73.9 | 23.4 | 5.35 | 69.1 |
| + fine-tuning (random) | 41.8 | 5.13 | 70.9 | 22.0 | 5.25 | 69.9 |
| + fine-tuning (reliable) | 48.5 | 5.12 | 70.5 | 28.6 | 5.24 | 66.9 |
| + fine-tuning (random + rectified) | 49.2 | 5.13 | 72.0 | 32.0 | 5.05 | 64.5 |
| + fine-tuning (reliable + rectified) | **51.3** | 5.13 | 71.3 | 36.6 | 5.06 | 66.7 |
| + LMD (Lian et al., 2023) [1] | 35.8 | 4.65 | 49.4 | **51.9** | 4.77 | 44.2 |
| + MultiDiffusion (Bar-Tal et al., 2023) [2] | 29.2 | 4.40 | 36.2 | 51.4 | 4.19 | 39.6 |
| + Ranni (Feng et al., 2024b) | 50.7 | 4.43 | 46.6 | 35.5 | 4.38 | 28.4 |

self-generated data produced with random seeds, using CogVLM2's responses to rectify incorrect text prompts (details in Appendix A.3.1). Our findings reveal that fine-tuning on rectified data does not consistently outperform our method, despite containing fewer noisy prompts. Moreover, applying rectification to data generated with reliable seeds further amplifies improvement. These results suggest a distinct source of enhancement and highlights the importance of image data with more reliable object arrangements.

**Human Evaluation.** To validate the reliability of GPT-4o for this task, we conducted a human evaluation study presented in Table 3. The results demonstrate strong correlation between GPT-4o assessments and human judgments across all metrics, and both evaluation results consistently demonstrate substantial improvements in our best fine-tuned models.

### 5.1.2 QUALITATIVE RESULTS

Figure 7 presents a comparative analysis of our methods against baselines and the state-of-the-art inference-time counterpart Ranni (Feng et al., 2024b). Our analysis includes the proposed sampling strategy using reliable seeds and the fine-tuning strategy using rectified self-generated data produced with reliable seeds. The results reveal that pretrained models often generate image arrangements that are unlikely to accurately render the specified quantity (e.g, "six eggs") or omit required objects (e.g, "a dove on top of a volleyball" and "a butterfly on the left of a bird"). Our fine-tuning strategy effectively addresses these issues and leads to more feasible image arrangements overall. In contrast, the baseline fine-tuning strategy using self-generated data produced with random seeds shows limited impact on image arrangements. Additionally, Ranni exhibits a tendency to generate visual artifacts, particularly rendering the desired objects in a bad shape. Even when the specified spatial relations are correctly depicted (e.g., "a butterfly on the left of a bird"), the generated images from Ranni often suffers from low visual quality and/or irrelevant background. We provide additional qualitative comparisons in Appendix A.7.

---

[1]LMD and MultiDiffusion were evaluated on the 512 × 512 version of Stable Diffusion 2.1 due to the difficulty of implementation on the 768 × 768 version. We re-implemented our method on the 512 × 512 version and report results in Appendix A.6.1.

[2]We used the same layouts produced by LMD as the layout input for MultiDiffusion.

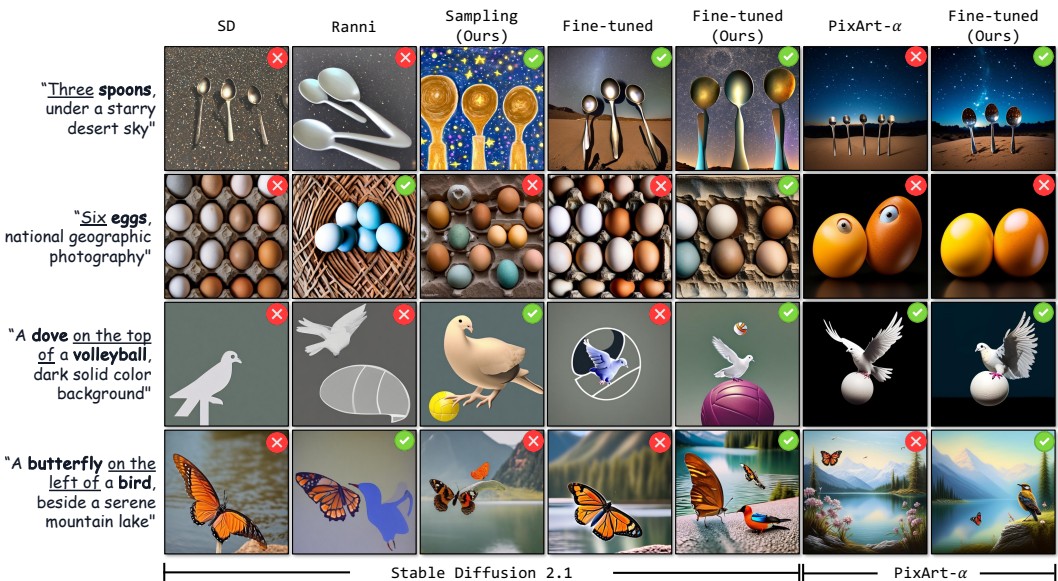

Figure 7: **Qualitative comparison** of different methods with various text prompts. **SD** represents the pre-trained Stable Diffusion 2.1. **Sampling (Ours)** represents the proposed sampling strategy using reliable seeds. **Fine-tuned** represents the baseline fine-tuning using data generated with random seeds. **Fine-tuned (Ours)** represents fine-tuning on rectified data generated with reliable seeds. For each row, images were generated with the same seed except for "Sampling (Ours)".

## 5.2 OUTPUT DIVERSITY AND IMAGE QUALITY

To assess the impact of our methods on image quality and output diversity, we employ two metrics: Aesthetic scores using the Aesthetic Predictor v2.5 (Christoph Schuhmann, 2022; ap2, 2024), and recall (Kynkäänniemi et al., 2019). We generated reference sets of 1,800 images for numerical composition and 1,920 images for spatial composition, using random seeds and text prompts from the test set. Recall, serving as a proxy for diversity, represents the proportion of reference images that fall within the manifold of the distribution of generated images being evaluated. Higher recall indicates greater diversity in the generated output. We compute recall on images generated by each method using the same text prompts but different random seeds.

For comparison, we evaluated two inference-time methods: MultiDiffusion and LLM-grounded Diffusion (LMD). The former achieves layout-to-image generation by fusing multiple diffusion paths and requires layout inputs. LMD mitigates the difficulty of obtaining layout inputs by employing LLMs to produce bounding boxes and performs "layout-grounded diffusion" with attention control.

Table 4 presents the aesthetic scores and recall for methods using Stable Diffusion 2.1. Our seed-based sampling strategy shows a slight drop in recall without decrease in aesthetic scores. All fine-tuning methods affect the aesthetic score and recall to a moderate extent. In contrast, LMD and MultiDiffusion exhibit more severe decreases in both aesthetic score and recall. This degradation may be attributed to visual artifacts introduced by the fusion of diffusion paths and attention control, as well as the limited diversity in LLM-produced layouts.

## 6 CONCLUSION

We explored the impact of initial seeds on compositional text-to-image generation and proposed a seed mining strategy to improve both numerical and spatial composition. Our analyses reveal that some initial random seeds lead to significantly more accurate composition, suggesting that inference-time scaling through optimized noise selection can enhance performance without retraining the model. Simply selecting reliable seeds improves generation quality, and additionally, it enables a fine-tuning approach based on self-generated reliable data, further enhancing compositional consistency. Future work could explore generalizing the seed mining strategy to other generative tasks, assessing its robustness as a broader inference-time scaling technique.

## 7 ACKNOWLEDGEMENT

This work was partially funded by the Swiss National Science Foundation via Project 200020_214878.

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

## A APPENDIX

### A.1 DATASET FOR TEXT-TO-IMAGE COMPOSITION - COMP90

#### A.1.1 OBJECTS

As presented in Table 5, we collected 90 distinct categories of common objects in life as our object dataset. We designated 60 for training and 30 for testing, namely, text prompts for evaluation do not contain any categories from the training set, and vice versa.

Table 5: Object categories in our dataset Comp90.

| Index | Object | Split | Index | Object | Split |
|-------|--------|-------|-------|--------|-------|
| 1 | Apple | Train | 46 | Koala | Train |
| 2 | Fish | Train | 47 | Bulb | Train |
| 3 | Envelope | Train | 48 | Orange | Train |
| 4 | Flamingo | Train | 49 | Gorilla | Train |
| 5 | Laptop | Train | 50 | Hamburger | Train |
| 6 | Camera | Train | 51 | Strawberry | Train |
| 7 | Carrot | Train | 52 | Owl | Train |
| 8 | Dolphin | Train | 53 | Pineapple | Train |
| 9 | Marble | Train | 54 | Pumpkin | Train |
| 10 | Snail | Train | 55 | Bottle | Train |
| 11 | Teapot | Train | 56 | Tree | Train |
| 12 | Coin | Train | 57 | Umbrella | Train |
| 13 | Monkey | Train | 58 | Vase | Train |
| 14 | Otter | Train | 59 | Whale | Train |
| 15 | Parrot | Train | 60 | Mushroom | Train |
| 16 | Dice | Train | 61 | Octopus | Test |
| 17 | Duck | Train | 62 | Unicorn | Test |
| 18 | Lemon | Train | 63 | Ant | Test |
| 19 | Mug | Train | 64 | Basketball | Test |
| 20 | Candle | Train | 65 | Cup | Test |
| 21 | Motorcycle | Train | 66 | Spoon | Test |
| 22 | Tent | Train | 67 | Tiger | Test |
| 23 | Horse | Train | 68 | Penguin | Test |
| 24 | Star | Train | 69 | Rabbit | Test |
| 25 | Turtle | Train | 70 | Dog | Test |
| 26 | Watermelon | Train | 71 | Elephant | Test |
| 27 | Zucchini | Train | 72 | Hat | Test |
| 28 | Notebook | Train | 73 | Airplane | Test |
| 29 | Boat | Train | 74 | Basket | Test |
| 30 | Raccoon | Train | 75 | Ukulele | Test |
| 31 | Phone | Train | 76 | Volleyball | Test |
| 32 | Pyramid | Train | 77 | Watch | Test |
| 33 | Car | Train | 78 | Feather | Test |
| 34 | House | Train | 79 | Pearl | Test |
| 35 | Windmill | Train | 80 | Clam | Test |
| 36 | Jellyfish | Train | 81 | Drone | Test |
| 37 | Guitar | Train | 82 | Bird | Test |
| 38 | Cat | Train | 83 | Egg | Test |
| 39 | Kangaroo | Train | 84 | Bowl | Test |
| 40 | Knife | Train | 85 | Chair | Test |
| 41 | Pillow | Train | 86 | Table | Test |
| 42 | Bus | Train | 87 | Dove | Test |
| 43 | Clock | Train | 88 | Crow | Test |
| 44 | Brush | Train | 89 | Panda | Test |
| 45 | Flower | Train | 90 | Butterfly | Test |

### A.1.2 BACKGROUND SETTINGS

To compose diversified text prompts, we hand-crafted 12 different background settings, listed in Table 6. We reserve 8 of them for training and 4 of them for testing.

Table 6: Hand-crafted background settings for text prompts.

| Index | Setting | Split |
|---|---|---|
| 1 | In an old European town | Train |
| 2 | On a snowy mountain | Train |
| 3 | On a rocky alien planet | Train |
| 4 | In a Swiss countryside | Train |
| 5 | Against the backdrop of a vibrant sunset | Train |
| 6 | In a misty jungle forest | Train |
| 7 | Beneath a shimmering aurora borealis | Train |
| 8 | On a sunny beach | Train |
| 9 | Dark solid color background | Test |
| 10 | Under a starry desert sky | Test |
| 11 | Beside a serene mountain lake | Test |
| 12 | National geographic photography | Test |

Table 7: Text prompt examples from Comp90.

| Text Prompt | Composition Type | Split |
|---|---|---|
| Six apples, against the backdrop of a vibrant sunset | Numerical | Train |
| Four fish, on a rocky alien planet | Numerical | Train |
| Two whales, on a snowy mountain | Numerical | Train |
| A dice on the top of a monkey, in an old European town | Spatial | Train |
| A lemon under a watermelon, in a Swiss countryside | Spatial | Train |
| An otter on the left of a hamburger, in a misty jungle forest | Spatial | Train |
| Three unicorns, under a starry desert sky | Numerical | Test |
| Six butterflies, national geographic photography | Numerical | Test |
| Five elephants, dark solid color background | Numerical | Test |
| A feather on the top of a panda, dark solid color background | Spatial | Test |
| A basketball on the right of a rabbit, under a starry desert sky | Spatial | Test |
| A volleyball on the top of a basketball, beside a serene mountain lake | Spatial | Test |

### A.1.3 PROMPTS

As described in Section 4.1, we produced prompts in the format "quantity object category, background setting" for the numerical dataset and "object category 1 spatial relation object category 2, background setting" for the spatial dataset. In the end, we obtained 2,400 / 600 prompts for training / testing for the numerical dataset, and 2,560 / 640 for the spatial dataset. We showcased some of the prompts in Table 7.

We filtered out prompts containing unreasonable compositions like "a table on the top of a bowl". To do so, we used GPT-4o (Achiam et al., 2023; OpenAI) with the following prompt: *"Here are some scenes focused on the spatial relation '{spatial relation}'. Now analyze each of them about if the scene is logical and answer in the following format: <scene> | <logical_or_not> | <very_brief_justification>".*

### A.1.4 TEXT PROMPTS USED FOR FIGURE 2

In the preliminary experiment that we demonstrated in Figure 2, we generated images using 64 text prompts composed of 8 object categories and 8 background settings in the format "Four object category, background setting". The 8 categories chosen for this experiment are "apple", "orange", "coin", "umbrella", "bottle", "dog", "cat", "boat". The 8 background settings are those in the training dataset, displayed in Table 6.

## A.2 TRAINING DETAILS

For all experiments requiring fine-tuning with reliable seeds, we used the top-3 best-performing seeds based on the results of seed mining. We fine-tuned Stable Diffusion 2.1 for 5,000 iterations using two NVIDIA A100 GPUs (each with 82 GB VRAM), which took 8 hours per run. We fine-tuned PixArt-$\alpha$ for 2,000 iterations using a single A100 GPU, with each procedure taking 2 hours.

For fine-tuning Stable Diffusion 2.1, we set the batch size to 16 per GPU and the number of gradient accumulation steps to 4, resulting in an effective batch size of 128. Consequently, we used a scaled learning rate of $1.28 \times 10^{-4} = 10^{-6} \times 2 \times 16 \times 4$. During fine-tuning, all parameters were frozen except for those in the Q, K projection layers of attention modules, excluding those in the first down-sampling block and the last up-sampling block in the U-Net. All other training-related configurations were set to the default values in the open-source package Diffusers that we employed.

For fine-tuning PixArt-$\alpha$, we set the batch size to 64 and the learning rate to $2 \times 10^{-5}$, with gradient clipping set to 0.01. All other training-related configurations were set to the default values in the official implementation.

## A.3 UTILIZATION OF OFF-THE-SHELF VISION LANGUAGE MODELS

### A.3.1 COGVLM2

CogVLM2 (Wang et al., 2023; Hong et al., 2024) is a GPT4V-level open-source visual language model based on Llama3-8B. We employed CogVLM2 throughout our work to predict the actual quantity or spatial relation in generated images to avoid manual labor.

- **Numerical composition**. For images containing numerical composition, we queried CogVLM2 with the prompt: "*Answer in one sentence: How many* {objects} *are in this image?*" We then extracted and transformed the quantity information from its responses into numeric words (e.g., "zero", "one", ..., "ten", "numerous"). Responses such as "there are too many to count" were translated to "numerous". We also classified predicted quantities larger than 19 as "numerous" due to CogVLM2's limitations in precise counting at higher numbers. For rectification, we replaced the numeric word in the original prompts with the predicted and translated word.

- **Spatial composition**. For images containing spatial composition, we employed a two-step querying process: (1) "*Describe the positions of the objects in the image in one sentence*" and then (2) "*Answer with yes or no: Is there a* {object 1} *positioned* {spatial relation} *a* {object 2} *in the image?*". We then searched for "yes" or "no" in the response to the second query to determine whether CogVLM2 predicts that the spatial relation aligns with the image. For rectification, due to the complexity of automatically extracting spatial relations from CogVLM2's responses, we replaced the entire text prompt with its description if the second response was "no".

In Figure 8, we showcase examples of using CogVLM2 to determine the number of objects in images generated by Stable Diffusion. These images are usually considered more challenging due to the object occlusions. Despite the complexities, CogVLM2 consistently provides accurate answers. Additionally, its output follows a consistent and standardized format, facilitating straightforward extraction of quantity information.

To demonstrate that our fine-tuned model does not overfit to generating only images with non-overlapping layouts, in Figure 9, we present examples of images generated by the fine-tuned Stable Diffusion model.

**Potential Biases** While CogVLM2 has shown strong performance in our evaluations, it is important to consider potential limitations and biases. The model may not perform well on objects that were not sufficiently represented in its training data, which could result in inaccuracies in assessing image composition. Similarly, cases with very large numbers of objects may pose challenges, as the model might struggle to correctly interpret complex spatial relationships. Additionally, linguistic biases may arise when dealing with languages that are not adequately represented in the training dataset, leading to suboptimal performance in vision-language tasks.

To address these limitations, potential approaches could include integrating a human evaluation process to validate automated assessments, especially for edge cases. Another direction is the use of

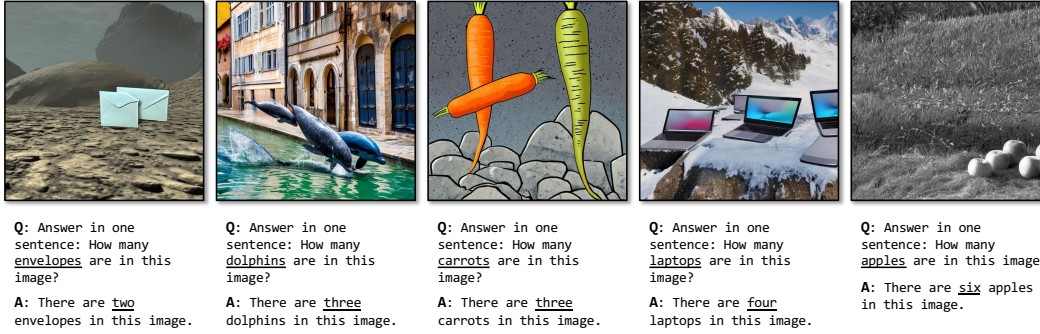

Figure 8: **Examples of using CogVLM2 to determine the object quantity. Q**: the text used for prompting CogVLM2. **A**: the output of CogVLM2.

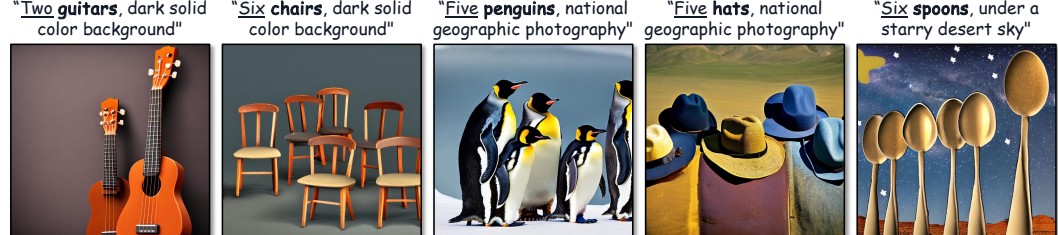

Figure 9: **Examples of generated images with object occlusions,** produced by our fine-tuned Stable Diffusion model.

ensemble models, where the outputs of multiple vision-language models are compared and aggregated through a voting mechanism. Such approaches could enhance reliability, reduce individual model biases, and improve performance in diverse and challenging scenarios.

### A.3.2 GPT-4O

GPT-4o (OpenAI; Achiam et al., 2023) is a state-of-the-art multi-modal model with advanced vision understanding capabilities. We employed it throughout our evaluations to predict the actual quantity or spatial relation in generated images. For images containing numerical composition, we queried GPT-4o with the same prompts used with CogVLM2 and extracted the quantity information in a similar manner. We classified predicted quantities larger than 19 as 19 due to the limited capability in precise counting at higher numbers and to establish an upper limit for mean absolute error (MAE) computation. For images containing spatial composition, we employed a two-step querying process. (1) "*Is there any* {object 1} *in the image? Is there any* {object 2}*? What is their spatial relation?*" and then (2) "*Based on your description, answer with yes or no: Is there a* {object 1} {spatial relation} *a* {object 2} *in this image?*". We then searched for "yes" or "no" in the second response to determine whether the generated image correctly rendered the spatial relation.

### A.4 SAMPLING FINE-TUNED MODELS WITH RELIABLE SEEDS

While our primary goal is to achieve seed-independent enhancement and enable the use of fine-tuned models with random seeds in standard applications, Table 8 demonstrates that sampling fine-tuned models with reliable seeds brings further performance enhancements across all metrics for both Stable Diffusion 2.1 and PixArt-α. Notably, these improvements are less pronounced compared to the enhancements reliable seeds bring to pre-trained models. This reduced impact is expected, as fine-tuning on data generated with reliable seeds has already incorporated some of the beneficial behaviors and reliable object arrangements associated with these seeds, leaving less room for further improvement when sampling. These findings suggest potential for further optimization in certain applicable scenarios.

Table 8: **Using fine-tuned models with reliable seeds yield further performance gains.**

| Method | Numerical | | Spatial |
| | Acc↑ | MAE↓ | Acc↑ |
|---|---|---|---|
| Stable Diffusion 2.1 | 37.5 | 1.39 | 17.8 |
| + sampling with reliable seeds | 43.0 | 1.29 | 23.4 |
| + fine-tuning (reliable + rectified) | 51.3 | 0.80 | 36.6 |
| + fine-tuning (reliable + rectified) + sampling with reliable seeds | **56.2** | **0.65** | **37.7** |
| PixArt-$\alpha$ | 34.3 | 1.74 | 22.7 |
| + sampling with reliable seeds | 40.3 | 1.60 | 25.6 |
| + fine-tuning (reliable + rectified) | 41.2 | 1.38 | 27.2 |
| + fine-tuning (reliable + rectified) + sampling with reliable seeds | **43.0** | **1.29** | **28.4** |

Table 9: **Comparison of different parameter selections for fine-tuning on the aesthetic score and the recall.**

| Method | Acc↑ | Aes↑ | Rec↑ |
|---|---|---|---|
| Stable Diffusion 2.1 | 37.5 | 5.19 | 76.7 |
| + fine-tuning (all parameters) | 23.8 (-13.7) | 4.12 (-1.07) | 33.1 (-43.6) |
| + fine-tuning (Q, K, V in self + cross attentions) | 54.7 (+17.2) | 5.03 (-0.16) | 54.7 (-22.0) |
| + fine-tuning (Q, K in cross attentions) | 48.5 (+11.0) | 5.15 (-0.04) | 75.0 (-1.7) |
| + fine-tuning (Q, K in self attentions) | 48.3 (+10.8) | 5.12 (-0.07) | 73.2 (-3.5) |
| + fine-tuning (Q, K in self + cross attentions) | 51.3 (+13.8) | 5.13 (-0.06) | 71.3 (-5.4) |

## A.5 ABLATION STUDIES

### A.5.1 OPTIMIZING PARAMETER SELECTION FOR FINE-TUNING

Fine-tuning a model on self-generated data requires careful consideration to avoid exacerbating visual biases present in the generated data Shumailov et al. (2023). Our goal was to enhance the model's compositional generation capabilities while preserving its overall performance. To achieve this, we focused on fine-tuning only the most relevant model components. We evaluated five configurations of trainable parameters:

1. All parameters
2. Q, K, V projection layers in all attention modules
3. Q, K projection layers in all cross-attention modules
4. Q, K projection layers in all self-attention modules
5. Q, K projection layers in all attention modules

For each configuration, we assessed output accuracy, aesthetic scores, and recall. Table 9 presents our findings. Our results indicate that configuration 5 (Q, K projection layers in all attention modules) yields substantial accuracy improvement while moderately impacting aesthetic scores and recall. This approach effectively balances enhanced compositional generation with the preservation of the model's original capabilities.

### A.5.2 NUMBER OF RELIABLE SEEDS

Table 10 reports the results of using Stable Diffusion 2.1 to generate images for numerical composition, using different numbers of top seeds obtained from seed mining. The results reveal that our sampling strategy maintains consistent aesthetic scores across various values of $k$, indicating that the visual quality of the generated images is not compromised as we increase the number of reliable seeds. Interestingly, we observe that larger $k$ values lead to higher recall, suggesting an increase in the diversity of generated images. This trend is logical, as a larger $k$ provides more seeds for use, resulting in a wider range of possible image arrangements. Even at $k = 50$, which represents half the number of all candidate seeds, we still see a substantial improvement in accuracy from 37.5% to 40.8%.

Table 10: **Comparison of different methods on the aesthetic score and the recall.**

| Method | Acc↑ | Aes↑ | Rec↑ |
|---|---|---|---|
| Stable Diffusion 2.1 | 37.5 | 5.19 | 76.7 |
| + sampling with top-1 reliable seeds | 42.8 | 5.28 | 74.1 |
| + sampling with top-3 reliable seeds | 43.0 | 5.23 | 73.9 |
| + sampling with top-10 reliable seeds | 37.8 | 5.19 | 75.4 |
| + sampling with top-20 reliable seeds | 42.0 | 5.16 | 76.0 |
| + sampling with top-30 reliable seeds | 41.2 | 5.13 | 76.6 |
| + sampling with top-50 reliable seeds | 40.8 | 5.22 | 76.1 |

Table 11: **Comparison of different methods on the aesthetic score and the recall.** Our method significantly improves the accuracy, while coming at a much lower loss of aesthetic score and recall than the state of the art (LMD and MultiDiffusion).

| Method | Numerical | | | Spatial | | |
|---|---|---|---|---|---|---|
| | Acc↑ | Aes↑ | Rec↑ | Acc↑ | Aes↑ | Rec↑ |
| Stable Diffusion 2.1 ($512 \times 512$) | 34.0 | 4.39 | 71.9 | 16.9 | 4.55 | **74.7** |
| + sampling with reliable seeds | 37.0 | 4.39 | **76.7** | 18.8 | 4.44 | 70.9 |
| + fine-tuning (random) | 33.7 | 3.98 | 70.4 | 19.1 | 4.19 | 66.4 |
| + fine-tuning (reliable) | 37.3 | 3.93 | 72.8 | 23.4 | 4.11 | 69.7 |
| + fine-tuning (random + rectified) | 40.7 | 3.94 | 70.3 | 25.2 | 3.93 | 62.4 |
| + fine-tuning (reliable + rectified) | **43.2** | 3.99 | 71.2 | 25.6 | 3.86 | 62.0 |
| + LMD (Lian et al., 2023) | 35.8 | **4.65** | 49.4 | **51.9** | **4.77** | 44.2 |
| + MultiDiffusion (Bar-Tal et al., 2023) [3] | 29.2 | 4.40 | 36.2 | 51.4 | 4.19 | 39.6 |

## A.6 ADDITIONAL EXPERIMENTS

### A.6.1 EVALUATION ON STABLE DIFFUSION $512 \times 512$

We have re-implemented our method using SD 2.1 $512 \times 512$ to ensure a fair comparison with LMD and MultiDiffusion. The results are shown in Table 11, indicating consistent performance trends with our approach continuing to excel in numerical composition and recall, while LMD and Multi-Diffusion show strengths in spatial composition. Additionally, we provide qualitative comparisons in Appendix A.7, which demonstrate the evident visual artifacts in images generated by LMD and MultiDiffusion although the aesthetic score metric fails to reflect them.

### A.6.2 EVALUATION ON MULTIPLE-CATEGORY NUMERICAL COMPOSITION

We constructed a specialized evaluation dataset to assess model performance on generating images containing specified quantities of multiple object categories. The dataset consists of 600 prompts, such as "*An ant and a basketball, dark solid color background*" and "*Five crows and a butterfly, beside a serene mountain lake*". Specifically, they are composed by 4 background settings (see Table 6, 10 pairs of categories (see Table 13), and 15 numerical requirements, which are $(1, 1), (1, 2), (2, 1), \cdots, (3, 3), (4, 2), (5, 1)$.

As shown in Table 12, we evaluate our sampling method and fine-tuned models, along with Ranni (Feng et al., 2024b) on this dataset. Note that we use the same reliable seeds and fine-tuned models obtained originally for the single-category numerical composition task. Specifically, for sampling with reliable seeds, we use the seeds for the quantity $x + y$ if the text prompt is "*x cups and y spoons.*" for example. The results indicate that our method generalizes well to multi-category scenarios and consistently outperforms baseline methods, even without conducting specifically designed seed mining. We provide qualitative comparisons in Figure 11.

---

[3]We used the same layouts produced by LMD as the layout input for MultiDiffusion.

Table 12: **Evaluation on Multiple-Category Numerical Composition.**

| Method | Acc↑ | Aes↑ | Rec↑ |
|---|---|---|---|
| Stable Diffusion 2.1 $(768 \times 768)$ | 10.0 | **5.06** | **71.4** |
| + sampling with reliable seeds | 11.5 | 5.08 | 68.6 |
| + fine-tuning (reliable + rectified) | **15.7** | 4.91 | 61.6 |
| + Ranni (Feng et al., 2024b) | 12.3 | 4.46 | 41.8 |
| PixArt-$\alpha$ | 12.8 | 5.10 | **81.7** |
| + sampling with reliable seeds | 14.8 | 4.88 | 64.7 |
| + fine-tuning (reliable + rectified) | **16.5** | 4.86 | 67.6 |

Table 13: Category pairs selected for multiple-category numerical composition.

| Index | Category 1 | Category 2 |
|---|---|---|
| 1 | Ant | Basketball |
| 2 | Cup | Spoon |
| 3 | Tiger | Penguin |
| 4 | Rabbit | Dog |
| 5 | Hat | Basket |
| 6 | Ukulele | Volleyball |
| 7 | Bird | Egg |
| 8 | Bowl | Dove |
| 9 | Chair | Panda |
| 10 | Crow | Butterfly |

### A.6.3 EVALUATION ON NUMERICAL COMPOSITION OF OUT-OF-SCOPE QUANTITIES

We created an additional numerical dataset for evaluating model performance on generating images containing more than six objects. Specifically, we developed this dataset in the same way as described in Section 4.1 and Appendix A.1, but only use "seven" and "eight" as the instructed quantity. In the end, we obtained 300 prompts.

As shown in Table 14, we evaluate our fine-tuned models on this dataset. Note that we did not re-train our method to tailor to this scenario but used the existing fine-tuned model without any modifications. The results indicate that our method generalizes very well to the out-of-scope scenarios and consistently outperforms baseline methods, even without conducting specifically designed seed mining.

### A.7 ADDITIONAL QUALITATIVE COMPARISON

We provide qualitative comparisons in Figure 10. Compared to Ranni (Feng et al., 2024b), LMD (Wang et al., 2023), and MultiDiffusion (Bar-Tal et al., 2023), our method generates images that are more visually similar to the original ones, and ours adhere to the backgrounds in the prompts significantly better. Additionally, it can be observed that there are often evident visual artifacts in

Table 14: **Comparison of different methods on numerical composition of out-of-scope quantities.** Results for different desired quantities are displayed separately, and "Avg" denotes the average of all quantities. **Bold** denotes the best value in each column.

| | Avg | | 7 | | 8 | |
|---|---|---|---|---|---|---|
| Method | Acc↑ | MAE↓ | Acc↑ | MAE↓ | Acc↑ | MAE↓ |
| Stable Diffusion 2.1 | 8.7 | 3.27 | 8.3 | 2.85 | 9.2 | 3.68 |
| + fine-tuning (reliable + rectified) | **16.7** | **1.97** | **10.0** | **1.79** | **23.3** | **2.15** |
| PixArt-$\alpha$ | 5.8 | 3.76 | 3.3 | 3.73 | **8.3** | 3.78 |
| + fine-tuning (reliable + rectified) | **8.3** | **3.21** | **9.2** | **3.02** | 7.5 | **3.40** |

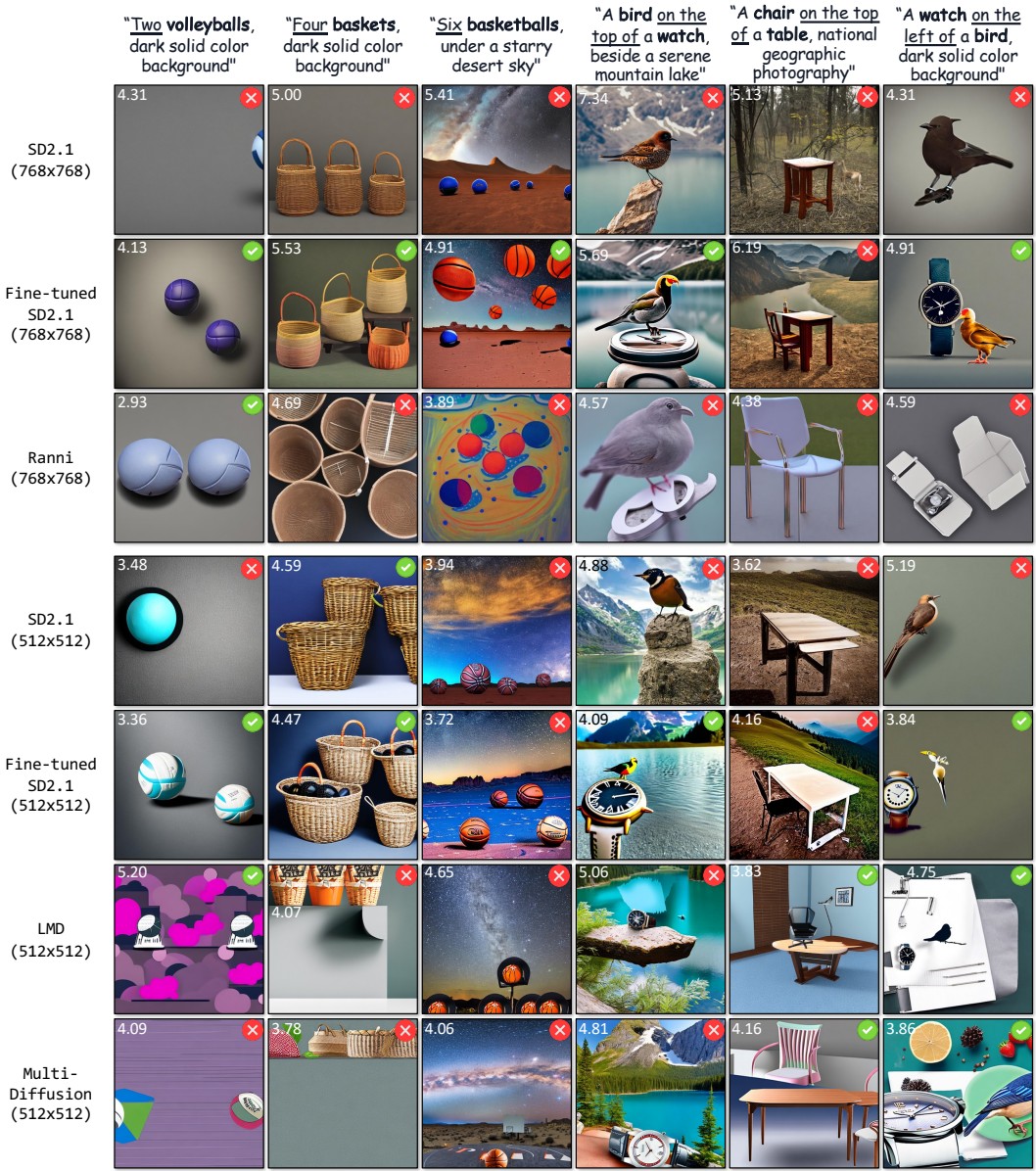

Figure 10: **Qualitative comparison** of different methods on different versions of Stable Diffusion 2.1. The aesthetic score (Christoph Schuhmann, 2022; ap2, 2024) is labeled at the top-left corner for each generated image.

images generated by LMD and MultiDiffusion although the aesthetic score metric fails to reflect them.

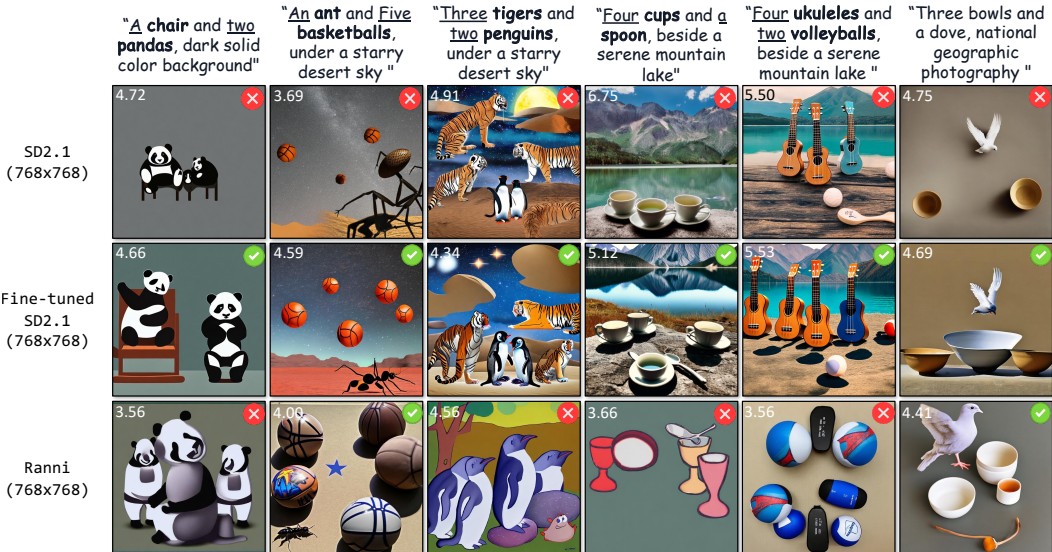

Figure 11: **Qualitative comparison** of different methods with text prompts for multiple-category numerical composition. The aesthetic score (Christoph Schuhmann, 2022; ap2, 2024) is labeled at the top-left corner for each generated image.

