# OpenReview forum: "Enhancing Compositional Text-to-Image Generation with Reliable Random Seeds"
_ICLR.cc/2025/Conference — ICLR 2025 Spotlight_

### Official Review · Reviewer_MsjD · 2024-10-30

**Soundness:** 3
**Presentation:** 3
**Contribution:** 3
**Rating:** 6
**Confidence:** 5

**Summary:**

This paper addresses the challenges faced by text-to-image models in handling compositional prompts, such as accurately rendering object quantities and spatial relations. It highlights the impact of initial random seeds on the arrangement and fidelity of generated images, proposing a method to improve model performance by identifying and leveraging “reliable seeds.” The paper’s main contributions include: 1) a generation strategy based on reliable seeds to reduce the need for manual annotations by automatically generating a high-quality dataset; 2) fine-tuning the model on self-generated reliable data to enhance numerical and spatial compositional accuracy; and 3) implementing a seed-based sampling strategy that improves generation accuracy without additional computation or training.

**Strengths:**

1. Provides a novel, data-efficient method to improve compositional accuracy in text-to-image generation by harnessing seed variability.
2. The automatic generation of a training dataset with reliable seeds reduces the labor-intensive process of manual annotation.
3. Extensive quantitative and qualitative evaluations demonstrate the approach’s effectiveness in improving both numerical and spatial compositional tasks across different models.

**Weaknesses:**

1. The reliance on selected seeds may limit the diversity of generated outputs, as increasing accuracy through reliable seeds could restrict the model’s range of variations.
2. There is no method presented for automatically selecting reliable seeds during inference, limiting the approach’s applicability to other models and use cases.
3. Potential decline in overall image generation quality when fine-tuning on self-generated data remains unexplored, especially concerning aesthetics and real-world accuracy.
4. The approach assumes that data generated with reliable seeds is of higher quality for model fine-tuning, but lacks empirical comparisons with real-world datasets or alternative high-quality sources.
5. Limited generalization testing to other diffusion models beyond Stable Diffusion and PixArt-α; therefore, the approach’s adaptability to diverse architectures is unclear.

**Questions:**

The citation format is slightly less standardized and the consistency of the references in the citation section should be ensured.

One key limitation is the lack of an automatic, inference-time method to select reliable seeds for generating accurate compositions. Would the authors consider developing a mechanism, such as a predictive model or algorithm, to dynamically choose reliable seeds based on prompt characteristics? This would significantly improve the model’s generalizability and practical use.

Fine-tuning on self-generated data inherently risks reducing image diversity or amplifying generation biases. Could the authors clarify how they ensured that this self-generated dataset maintains high quality compared to real-world or externally validated datasets? Additionally, what safeguards are in place to prevent potential degradation in image quality or unintended biases?

---

> ### Author Response · Authors · 2024-11-27
>
> We appreciate the reviewer's detailed feedback and their positive assessment of our method's novelty, efficiency and experimental validation. Below we address each point raised in the Weaknesses and Questions sections:
>
> > **The reliance on selected seeds may limit the diversity of generated outputs**
>
> Thank you for raising this important point, as it aligns closely with a key contribution in our paper. We specifically address concerns about limiting model diversity by adopting a novel fine-tuning scheme that modifies only the "Query" and "Key" projection layers in the attention modules. This approach is designed to preserve the model’s inherent diversity while improving accuracy (see Table 9 for more details). As shown in Table 4, our method achieves a Recall score of 71.3, which is remarkably close to the original model’s score of 76.7, indicating that diversity is largely maintained. In contrast, other methods, such as LMD (Recall = 49.4), MultiDiffusion (Recall = 36.2), and Ranni (Recall = 46.6), show significantly reduced diversity. Additionally, if diversity were not a major concern, our approach could achieve even higher accuracy, as demonstrated in Table 9.
>
>
> > **Lack of an automatic, inference-time method to select reliable seeds for generating accurate compositions. Will the authors consider developing an algorithm to dynamically choose seeds based on prompt characteristics?**
>
> Thank you for raising this important point. We would like to clarify that our seed mining method is indeed automatic via utilizing CogVLM. Perhaps the underlying concern here is the efficiency and practicality of applying this approach on-the-fly for a specific testing prompt. We acknowledge that, in its current implementation, the process can be somewhat slow for such use cases, as it involves generating hundreds of images to evaluate the reliability of a random seed for a given prompt. This being said, this approach remains feasible for more efficient approaches, such as one-step diffusion models, where the computational overhead is significantly reduced. Additionally, we find the reviewer’s suggestion of a direct mapping from text embeddings to reliable noise patterns highly intriguing. This could indeed streamline the seed selection process and improve efficiency. However, we believe that collecting the paired training data and training such a model would require substantial effort, making it an excellent avenue for future work. We are excited by the potential of this direction and would be happy to pursue it in follow-up research, building on the foundations established in this study. Thank you for highlighting this compelling opportunity.
>
>
> > **Potential decline in image quality is unexplored. Are there measures to prevent potential degradation in image quality or unintended biases? How do the authors ensure that the self-generated dataset maintains high quality compared to real-world or externally validated datasets?**
>
> Thank you for raising this concern. A drop in image quality is indeed a challenge for all methods aiming to improve model fidelity, including ours. However, as shown in Table 4, our method achieves a much smaller loss in aesthetic scores—a proxy for image quality—compared to state-of-the-art methods, while still improving accuracy, as can be seen from Table 4:
>
>  **Aesthetic Scores:** SD: 5.19, LMD: 4.65, MultiDiffusion: 4.40, Ranni: 4.43, Ours: 5.13.
>
> This improvement is largely due to our fine-tuning scheme, which modifies only a small part of the model, preserving the overall image quality. In contrast, as shown in Table 9, fine-tuning all model parameters significantly degrades image quality (Aesthetic Score: 4.12).
>
>
> > **The approach assumes that data generated with reliable seeds is of higher quality for model fine-tuning, but lacks empirical comparisons with real-world datasets.**
>
> We would like to clarify that our approach does not claim that synthetic data is inherently better than real-world data. Instead, our focus is on the economic and practical benefits of using self-generated data, which provides a cost-effective way to fine-tune models, especially when high-quality real-world datasets are scarce or difficult to obtain. While empirical comparisons with real-world datasets can be interesting, a direct apples-to-apples comparison is non-trivial since a sufficiently diverse set of high-quality real-world images for specific prompts, such as "four coins," is not always readily available or easy to assemble. If the reviewer has a specific real-world dataset in mind that could be suitable for comparison, we would be happy to explore it and provide further insights.

---

> > ### Author Response · Authors · 2024-11-27
> >
> > > **Limited generalization testing to other diffusion models beyond Stable Diffusion and PixArt-alpha. The method's adaptability is unclear**
> >
> >  Thank you for your comment. We must note that we test our proposed method on both Stable Diffusion (SD) and PixArt-alpha, while previous research has primarily focused on improving Stable Diffusion alone (LMD, MultiDifussion, and Ranni [1]). We chose these two models because they represent distinct architectures: Stable Diffusion, a U-Net based diffusion model, and PixArt-alpha, a Transformer-based diffusion model. We believe testing on these two models sufficiently demonstrates our method's ability to generalize across different generative architectures, ensuring its robustness and applicability.
> >
> >  [1] Feng, Yutong, et al. "Ranni: Taming text-to-image diffusion for accurate instruction following." Proceedings of the IEEE/CVF Conference on Computer Vision and Pattern Recognition. 2024.

---

### Official Review · Reviewer_uQGW · 2024-11-02

**Soundness:** 4
**Presentation:** 4
**Contribution:** 3
**Rating:** 8
**Confidence:** 4

**Summary:**

This paper tackles two main aspects of diffusion models: Numerical and spatial generations. The aim is to use the diffusion model as is without additional inputs such as layouts. First, reliable seeds are mined which produce correct results for the numerical and spatial generations. Then, these seeds are used to create a generative dataset and the model is fine-tuned on this dataset to improve performance.

**Strengths:**

1. The main advantage of this work is that no additional modules/trainable parameters need to be added to the diffusion model which incorporate layouts or bounding boxes like other works usually do.
2. Extensive experimentation is conducted to validate the reliable seeds hypothesis.
3. Once reliable seeds are mined, the authors have experimented with a broad spectrum of ways to use that to enhance the model's performance.

**Weaknesses:**

1. Baselines: Newer methods to accomplish this task have developed such as [1] after LMD, which show an improvement over LMD. This work should be compared with [1] instead of LMD to demonstrate the efficacy of this approach.

I have clubbed the other points in the questions section

---
[1] Feng, Yutong, et al. "Ranni: Taming text-to-image diffusion for accurate instruction following." Proceedings of the IEEE/CVF Conference on Computer Vision and Pattern Recognition. 2024.

**Questions:**

1. What are the numbers when compared to that of [1]?

---

2. Comparison against baselines: As 512x512 implementation of Stable Diffusion is used for LMD and Multi Diffusion, comparing it with 768x768 version becomes unfair. What are the numbers for Table 4 when using the 512x512 Stable diffusion of this method instead of the 768x768?

---

3. Mixture of objects for numerical compositions. All the results seem to display numerical compositions of a single object. How are the results when I compose multiple objects, such as "2 airplanes and 4 birds in the sky", and how do the baselines compare with this method for such cases?

---
---
I will reconsider my rating if these concerns are addressed.

Please correct me if you think I have misunderstood any aspect of the paper.

---

> ### Author Response · Authors · 2024-11-25
>
> We appreciate the reviewer's thoughtful feedback and their positive assessment of our method's simplicity, broad applicability, and experimental validation. Below we address each point raised in the Weaknesses and Questions sections:
>
> ---
>
> > **Comparison with Ranni [1].**
>
> We thank the reviewer for bringing Ranni to our attention. Similar to LMD, Ranni employs a two-stage generation process: text-to-panel and panel-to-image. We have conducted comparisons with Ranni (implemented on SD2.1 768x768) on our dataset. As shown in the updated Table 4, while Ranni achieves comparable numerical and spatial accuracy (50.7% vs. our 51.3%, and 35.5% vs. our 36.6%), it does so at a significantly higher cost to aesthetic quality and recall (Recall: 46.6 vs. our 71.3; Aesthetic Score: 4.43 vs. our 5.13). A qualitative comparison has also been added in Appendix A.7. We also would like to note that Ranni uses additional network structures for conditional input, requires extensive fine-tuning (40k steps vs. our 5k), and relies on a much larger dataset (50M samples vs. our 2.5k). These distinctions highlight the efficiency of our method in achieving competitive performance with fewer resources.
>
>
>
> | Method                                       | **Numerical**               |           |           | **Spatial**                 |           |           |
> |----------------------------------------------|-----------------------------|-----------|-----------|-----------------------------|-----------|-----------|
> |                                              | Acc ↑                       | Aes ↑    | Rec ↑    | Acc ↑                      | Aes ↑    | Rec ↑    |
> | Stable Diffusion 2.1 (768x768)               | 37.5                        | **5.19** | **76.7** | 17.8                        | **5.38** | **70.4** |
> | + sampling with reliable seeds               | 43.0                        | 5.23     | 73.9     | 23.4                        | 5.35     | 69.1     |
> | + fine-tuning (random)                       | 41.8                        | 5.13     | 70.9     | 22.0                        | 5.25     | 69.9     |
> | + fine-tuning (reliable)                     | 48.5                        | 5.12     | 70.5     | 28.6                        | 5.24     | 66.9     |
> | + fine-tuning (random + rectified)           | 49.2                        | 5.13     | 72.0     | 32.0                        | 5.05     | 64.5     |
> | + fine-tuning (reliable + rectified)         | **51.3**                    | 5.13     | 71.3     | 36.6                        | 5.06     | 66.7     |
> | + LMD [Lian et al., 2023]                    | 35.8                        | 4.65     | 49.4     | **51.9**                    | 4.77     | 44.2     |
> | + MultiDiffusion [Bar-Tal et al., 2023]      | 29.2                        | 4.40     | 36.2     | 51.4                        | 4.19     | 39.6     |
> | + Ranni [Feng et al., 2024b]                 | 50.7                        | 4.43     | 46.6     | 35.5                        | 4.38     | 28.4     |

---

> > ### Author Response · Authors · 2024-11-25
> >
> > > **LMD/MultiDiffusion output images at 512×512 while the method outputs images at 768x768?**
> >
> > Thank you for pointing this out. The main reason is that LMD and MultiDiffusion are not available for the 768×768 resolution models, and it is not trivial to reimplement these methods on higher-resolution models due to their complexity. To provide additional insights, we re-implemented our method on the 512×512 version of Stable Diffusion 2.1 and report the results in Appendix A.6.2, which show consistent performance trends; our approach continues to excel in numerical composition and recall, while LMD and MultiDiffusion show strengths in spatial composition.
> > We provide qualitative comparisons in Appendix A.7, which demonstrate the evident visual artifacts in images generated by LMD and MultiDiffusion although the aesthetic score metric fails to reflect them.
> > Additionally, it is worth noting that our proposed seed selection strategy is a distinct and orthogonal approach to inference-time methods such as LMD and MultiDiffusion. It is simple, easily adaptable, and could potentially be integrated with these methods in future work to leverage the strengths of both.
> >
> >
> >
> > | Method                                       | **Numerical**               |           |           | **Spatial**                 |           |           |
> > |----------------------------------------------|-----------------------------|-----------|-----------|-----------------------------|-----------|-----------|
> > |                                              | Acc ↑                      | Aes ↑    | Rec ↑    | Acc ↑                      | Aes ↑    | Rec ↑    |
> > | Stable Diffusion 2.1 (512×512)               | 34.0                        | 4.39     | 71.9     | 16.9                        | 4.55     | **74.7** |
> > | + sampling with reliable seeds               | 37.0                        | 4.39     | **76.7** | 18.8                        | 4.44     | 70.9     |
> > | + fine-tuning (random)                       | 33.7                        | 3.98     | 70.4     | 19.1                        | 4.19     | 66.4     |
> > | + fine-tuning (reliable)                     | 37.3                        | 3.93     | 72.8     | 23.4                        | 4.11     | 69.7     |
> > | + fine-tuning (random + rectified)           | 40.7                        | 3.94     | 70.3     | 25.2                        | 3.93     | 62.4     |
> > | + fine-tuning (reliable + rectified)         | **43.2**                    | 3.99     | 71.2     | 25.6                        | 3.86     | 62.0     |
> > | + LMD [Lian et al., 2023]                    | 35.8                        | **4.65** | 49.4     | **51.9**                    | **4.77** | 44.2     |
> > | + MultiDiffusion [Bar-Tal et al., 2023]      | 29.2                        | 4.40     | 36.2     | 51.4                        | 4.19     | 39.6     |
> >
> >
> >
> > > **How does our method perform in the case of multiple-category numerical composition?**
> >
> > Thank you for this interesting suggestion regarding multi-category scenarios (e.g., "2 airplanes and 4 birds"). These scenarios are significantly more challenging than single-category cases.
> > To test this scenario, we created a test set of 600 prompts, featuring 10 category pairs with diverse backgrounds and numerical instructions (see Appendix A.6.2).
> > Our evaluation in the table below shows that all existing methods, including ours, face difficulties in such conditions. Note that we did not re-train our method to tailor to this scenario but used the  same model fine-tuned for single-category scenarios without any modifications.
> > As can be seen in the table below, our approach achieves a 5.7% increase in accuracy compared to the original SD model and outperforms Ranni by 3.4%. It also improves Pixel-α by 3.7%.
> >
> > | Method                                    | Acc ↑  | Aes ↑  | Rec ↑  |
> > |-------------------------------------------|--------|--------|--------|
> > | Stable Diffusion 2.1 (768×768)            | 10.0   | **5.06**| **71.4**|
> > | + sampling with reliable seeds            | 11.5   | 5.08   | 68.6   |
> > | + fine-tuning (reliable + rectified)      | **15.7**| 4.91   | 61.6   |
> > | + Ranni [Feng et al., 2024]               | 12.3   | 4.46   | 41.8   |
> > |-------------------------------------------|--------|--------|--------|
> > | PixArt-α                                  | 12.8   | **5.10**| **81.7**|
> > | + sampling with reliable seeds            | 14.8   | 4.88   | 64.7   |
> > | + fine-tuning (reliable + rectified)      | **16.5**| 4.86   | 67.6   |
> >
> >
> >
> > [1] Feng, Yutong, et al. "Ranni: Taming text-to-image diffusion for accurate instruction following." Proceedings of the IEEE/CVF Conference on Computer Vision and Pattern Recognition. 2024.
> >
> > ---
> >
> > We hope these clarifications and additional experiments address the reviewer’s concerns. If further clarifications are needed, we are happy to provide them.

---

> > > ### Comment · Reviewer_uQGW · 2024-11-26
> > >
> > > Thank you. Could you also provide a qualitative comparison for the multiple category numerical composition case? Visual comparison against SD and Ranni should suffice (a similar diagram as in Fig 8 of A.7)

---

> > > ### Comment · Reviewer_uQGW · 2024-11-30
> > >
> > > The authors have addressed all my questions. Especially my concern about comparison against Ranni and the image resolutions issue. Further, they also included the visuals that I had asked for. I am raising my rating as I am satisfied with their response.
> > >
> > > However, I would strongly advice the authors to maintain two tables in the camera-ready version of the paper, one for 512x512 and 768x768 if it is difficult to implement LDM and Multi-Diffusion on 768x768. Two tables will make the comparisons fairer.

---

> > > > ### Author Response · Authors · 2024-11-30
> > > > **Thank you!**
> > > >
> > > > Thank you for your valuable feedback and for raising your rating! As per your suggestion, we will include two tables in the revised version. We sincerely appreciate your time and effort in reviewing our paper.

---

> ### Author Response · Authors · 2024-11-26
>
> Thank you for the suggestion! We have added another qualitative comparison between SD and Ranni for multiple-category numerical composition in Figure 11 in the new revision. If further comparisons are needed, we are happy to provide them.

---

### Official Review · Reviewer_ZEiT · 2024-11-03

**Soundness:** 3
**Presentation:** 3
**Contribution:** 3
**Rating:** 8
**Confidence:** 3

**Summary:**

In this paper, the author explores the random noise for the diffusion-based generation, especially for the text-to-image generation.

**Strengths:**

The idea is easy to follow.
The problem is well-designed.
The finding is interesting to many diffusion users.

**Weaknesses:**

1. Section 3.2 are not well-proved for the correlation. I am not convinced about the heatmap results.
- How do you decide the correct / incorrect in Figure 4?  Do this process bring the bias or prefrency over the distribution?
- Which layer is used for the heatmap? the output of diffusion model before VAE decoder?
- The four coins can be parallel or any position arrangement. So why the heatmap in Figure 4 is coincidently splited the 4 grids?

2. More compositional generation results and failure cases
How to generate the partially overlapped objects?
The samples showed are almost no overlapped.

3.  The definition of seed.
So you just fix the seed rather than the noise?
Everytime we will resample the noise according to the seed?
So why there will be a preferency over certain seed in Section3.3?

4. Minor problem
What are the "these images" in abstract? You may training images, which is collected by you? Please specify it.

5. Scalability to unseen prompts.
How about 7 or 8 objects?
How about the ``boundary'' or ``corner''?

**Questions:**

Please see the Weakness for details.

My main concern is as follows.
(1) Correctness is decided by the large model CogVLM2.  It will also lead to the bias, like preferring non-overlapping layout.
Finetuning makes the model overfitted.

(2) Fixed System Seed. You mean the input noise is also fixed? Actually, we fixed the input noise.

(3) Overlapped result is hard to generate.

---

> ### Author Response · Authors · 2024-11-27
>
> We appreciate the reviewer's thoughtful feedback and their positive assessment of our method's simplicity and usefulness. Below we address each point raised in the Weaknesses and Questions sections:
>
> > **How do we decide if an image is correctly generated in Figure 4? Does this process bring biases?**
>
> We utilized CogVLM2 to assess whether an image is correctly composed (details provided in Section 4.2 and Appendix A.3.1). This is an automatic approach, and we agree with the reviewer that a discussion about potential biases is necessary, which we have added in A.3.1. In our particular case, we have spent significant effort to manually verify the correctness of CogVLM2 and found no specific biases in the model. Additionally, we manually reviewed the 600 images generated to create Figure 4 (top) and found very few assessment errors from CogVLM2. The model performed well across various categories, backgrounds, and object arrangement patterns, including scenarios with overlapping objects.
>
>
> > **Does CogVLM2 result in biases, for example, preferency towards non-overlapping layout? Are the fine-tuned models overfitted to generating images with non-overlapping layouts?**
>
> We would like to clarify that many generated images with overlapping object layouts are correctly annotated by CogVLM2. To illustrate this, we provide several examples in Figure 8 of Appendix A.3.1 (in the revised paper). Figure 4 seems to highlight layouts with non-overlapping objects because the pre-trained Stable Diffusion and PixArt-alpha models tend to generate such images more frequently, not because of a bias in the evaluation model. Notably, our fine-tuned model does not overfit to a specific mode, as demonstrated by its significantly higher recall scores compared to all other baselines (Table 4).
> To illustrate that the fine-tuned models are not restricted to generating non-overlapping layouts, we showcase examples of images with object occlusions generated by our fine-tuned Stable Diffusion model in Figure 9 of Appendix A.3.1. We would like to point out that this is another strength of our method compared to layout-to-image methods such as LMD and MultiDiffusion, which rely on non-overlapping bounding boxes as input constraints.
>
>
>
>
> > **Which layer is used for the heatmap in Figure 4?**
>
> We follow the established convention in the field (e.g., [2], [3]) by computing the average of the attention maps across all timesteps and all cross-attention modules within the UNet model.
>
>
> > **The four coins can be parallel or any position arrangement. Why does the heatmap for "four coins" in Figure 4 is coincidently splited the 4 grids?**
>
>  We would like to clarify that this heatmap represents the behavior of the original pre-trained diffusion model. The pattern in Figure 4 occurs because the prompt "four coins" is most often composed when arranged in a 2x2 grid. This is likely influenced by the pre-training data, which contains many examples of "four" arranged in this way. However, this is not the only pattern the model can generate; other arrangements, such as parallel ones, as the reviewer suggests, are possible but less representative. The plot in Figure 4 does not emphasize these less frequent cases.
>
>
> > **Overlapped result is hard to generate?**
>
> We would like to note that our seed selection method does not favor any particular types of images, whether overlapping or non-overlapping. We do not impose any specific mechanism to check for this, and CogVLM does not bias toward any particular case, as far as we have observed. While controlling when the model generates overlapping versus non-overlapping objects could be useful, we believe that this goes beyond the scope of our current work.
>
> In general, how often and accurately a model generates overlapping objects depends largely on its training data. If the training data predominantly contains non-overlapping layouts, the model is less likely to generate accurate overlapping compositions. This is a common challenge with pre-trained generative models and is not specific to our method. Nevertheless, as shown in Figure 9, there are still many instances where overlapping objects are generated successfully.
>
> > **Do we control the noise by fixing the seed? Why is there a preferency over certain seeds in Section 3.3?**
>
> Yes, fixing the random seed effectively fixes the input noise for diffusion models. In principle, our paper highlights that certain noise patterns can be exploited to enhance model performance. In practice, we use the initial seeds as a straightforward and reproducible way to control the noise. Each seed uniquely determines a specific noise pattern, and with a fixed seed, the model consistently starts from the same deterministic noise rather than resampling. This leads to consistent positional arrangements and explains the observed preference for certain seeds in Section 3.3.

---

> ### Author Response · Authors · 2024-11-27
>
> > **What are the "these images" in abstract**
>
> Thank you for pointing this out. "These images" in the abstract means the generated images used for training. To make it clearer, we have changes "these images" to "these generated images".
>
>
> > **Scalability to out-of-scope prompts/tasks**
>
> Thank you for this interesting question. In principle, our method obtains the largest improvement when seed mining is conducted on the same task as that of the text prompts for testing. Nevertheless, our fine-tuned models demonstrate a good ability to generalize and improve the compositional accuracy for out-of-scope but related tasks (e.g., composing more than 6 objects in our case).
>
> To evaluate adaptability to unseen tasks, we tested our method on two new datasets: (1) numerical prompts with required quantities varying from 7 to 8, and (2) multi-category numerical compositions (such as "two cups and four spoons"). To test our method on multi-category numerical composition, we created a test set of 600 prompts, featuring 10 category pairs with diverse backgrounds and numerical instructions (see Appendix A.6.2).
>
> As shown in Appendix A.6.2, A.6.3 and the two tables below, our fine-tuned models extend their improvements to instructions that were never seen during seed mining or fine-tuning.
>
> **Table 1: Accuracy comparison when generating 7 and 8 objects**
>
> | Method                                  | Acc ↑ (Avg) |  MAE ↓ (Avg) |  Acc ↑ ( 7 ) |  MAE ↓ ( 7 ) |  Acc ↑ (8) | MAE ↓ (8) |
> |-----------------------------------------|---------------|---------------|-------------|-------------|-------------|-------------|
> | Stable Diffusion 2.1                    | 8.7           | 3.27          | 8.3         | 2.85        | 9.2         | 3.68        |
> | + fine-tuning (reliable + rectified)    | **16.7**      | **1.97**      | **10.0**    | **1.79**    | **23.3**    | **2.15**    |
> |-----------------------------------------|---------------|---------------|-------------|-------------|-------------|-------------|
> | PixArt-alpha                                | 5.8           | 3.76          | 3.3         | 3.73        | **8.3**     | 3.78        |
> | + fine-tuning (reliable + rectified)    | **8.3**       | **3.21**      | **9.2**     | **3.02**    | 7.5         | **3.40**    |
>
>
> **Table 2: Accuracy comparison for multi-category positional composition.**
>
> | Method                                    | Acc ↑  |
> |-------------------------------------------|--------|
> | Stable Diffusion 2.1                      | 10.0   |
> | + sampling with reliable seeds            | 11.5   |
> | + fine-tuning (reliable + rectified)      | **15.7**|
> |-------------------------------------------|--------|
> | PixArt-alpha                                  | 12.8   |
> | + sampling with reliable seeds            | 14.8   |
> | + fine-tuning (reliable + rectified)      | **16.5**|
>
>
> [1] Liu, Haotian, et al. "Visual instruction tuning." Advances in neural information processing systems 36 (2024).
>
> [2] Hertz, Amir, et al. "Prompt-to-prompt image editing with cross attention control." arXiv preprint arXiv:2208.01626 (2022).
>
> [3] Chen, Minghao, Iro Laina, and Andrea Vedaldi. "Training-free layout control with cross-attention guidance." Proceedings of the IEEE/CVF Winter Conference on Applications of Computer Vision. 2024.

---

> ### Comment · Reviewer_ZEiT · 2024-11-29
> **Thank you**
>
> The author has addressed my concerns, especially on occluded samples.
>
> The main idea is easy to follow.
>
> The technical contribution is weak, but it will interest most of our community if published.
>
> I raised my rating.

---

> > ### Author Response · Authors · 2024-11-29
> > **Thanks!**
> >
> > Thank you for your thoughtful feedback and for raising your rating. We appreciate your acknowledgment of the clarified points and the work's relevance to the community. We remain at your disposal for any further requests.

---

### Meta-Review · Area_Chair_VnnN · 2024-12-16

**Metareview:**

The authors present a seed-mining strategy for enhancing reliable compositional text-to-image generation. The rebuttal addressed several key concerns, leading two reviewers to raise their scores. Overall, the reviewers expressed positive sentiments about the paper, though some noted that the technical contributions could have been stronger. Additionally, the proposed approach demonstrates improved compositional capabilities, but this comes at the expense of diversity and image quality.

Despite these limitations, the paper provides compelling empirical results and contributes valuable insights to the field. Based on the reviewers' assessments and the discussion, the AC panel has decided to accept the paper.

**Additional Comments On Reviewer Discussion:**

Reviewer ZEiT's weakness comments were addressed by the reviewers and the score was increased.

Reviewer uQGW asked for comparison with Ranni (CVPR'24) approach and qualitative comparison for the multiple category numerical composition case, which were provided by the authors and the score was increased.

Reviewer MsjD is yet to participate in the discussion, though there were several important questions asked by the reviewer.

---

### Decision · Program_Chairs · 2025-01-22

Accept (Spotlight)